# Atmospheric Pollution and Thyroid Function of Pregnant Women in Athens, Greece: A Pilot Study

**DOI:** 10.3390/medsci8020019

**Published:** 2020-04-04

**Authors:** Ioannis Ilias, Ioannis Kakoulidis, Stefanos Togias, Stefanos Stergiotis, Aikaterini Michou, Anastasia Lekkou, Vasiliki Mastrodimou, Athina Pappa, Evangelia Venaki, Eftychia Koukkou

**Affiliations:** Department of Endocrinology, Diabetes and Metabolism, Elena Venizelou Hospital, 115 21 Athens, Greece; i_kakoulidis@yahoo.gr (I.K.); s.tog90@gmail.com (S.T.); stef_ster@hotmail.com (S.S.); katerina.michoy@yahoo.com (A.M.); anastasia.lk@gmail.com (A.L.); si-li-a@hotmail.com (V.M.); athpappa@gmail.com (A.P.); e_venaki@gmail.com (E.V.); ekoukkou@gmail.com (E.K.)

**Keywords:** pregnancy, thyrotropin, environmental air pollutants

## Abstract

Exposure to air pollution and, in particular, to nitrogen dioxide (NO_2_) or particulate pollutants less than 2.5 μm (PM2.5) or 10 μm (PM10) in diameter has been linked to thyroid (dys)function in pregnant women. We hypothesized that there may be a dose—effect relationship between air pollutants and thyroid function parameters. We retrospectively evaluated thyrotropin (TSH) in 293 women, NO_2_, PM2.5 and PM10 levels in Athens. All the women were diagnosed with hypothyroidism for the first time during their pregnancy. Exposure to air pollution for each woman was considered according to her place of residence. Statistical analysis of age, pregnancy weight change, and air pollutants versus TSH was performed with ordinary least squares regression (OLS-R) and quantile regression (Q-R). A positive correlation for logTSH and PM2.5(*r* = +0.13, *p* = 0.02) was found, using OLS-R. Further analysis with Q-R showed that each incremental unit increase (for the 10th to the 90th response quantile) in PM2.5 increased logTSH(±SE) between +0.029 (0.001) to +0.025 (0.001) mIU/L (*p* < 0.01). The other parameters and pollutants (PM10 and NO_2_) had no significant effect on TSH. Our results indeed show a dose—response relationship between PM2.5 and TSH. The mechanisms involved in the pathophysiological effects of atmospheric pollutants, in particular PM2.5, are being investigated.

## 1. Introduction

The offspring of pregnant women that have been exposed to air pollutants show structural brain alterations, reduced executive function and behavioral problems. The tentative mechanisms for these problems may be oxidative stress, neuroinflammation or changes in the hypothalamic-pituitary-adrenal axis [1]. Thyroid hormones are crucial in the development of the fetal brain. The fetal thyroid gland is fully functional after mid-pregnancy, thus undiagnosed or less than optimally treated thyroid insufficiency early on in pregnancy adversely affects the development the offspring. Studies have shown a link between pollutants (polycyclic aromatic hydrocarbons) and maternal thyroid function [1]. Particulate air pollutants are prevalent and harmful to humans (as assessed versus mortality, morbidity or cardiovascular disease) [2]; however, data on whether exposure to particulate air pollutants can also disrupt thyroid function in pregnancy are limited. Recently, exposure to ambient air pollution (and more in detail to nitrogen dioxide-NO_2_ or particulate air pollutants with a diameter of less than 2.5 micrometers-PM2.5 or 10 micrometers-PM10) has been linked to thyroid (dys)function in pregnant women (and/or their fetuses/neonates) [1,3,4,5]. The largest-scale study was based on satellite-assessed pollution measurements, whereas smaller scale studies were done with near-ground level air pollution assessments. These studies were honed on women living in northern Europe (*n* = 6472, the Netherlands [4]) or lower latitudes (*n* = 11,927, in China, Spain, USA or Greece [1,3,5]). Close evaluation of the reported results showed differences among Spain, USA and Greece (regarding the latter, results stemmed from a study in a provincial town and suburban/rural settings in the island of Crete). Nutritional (p.ex. regarding iodine deficiency) and environmental (p.ex. regarding type of pollutants exposed to) differences do exist among these populations [6,7,8,9,10,11,12,13,14,15,16,17,18,19]. The environmental degradation, which has occurred in recent years in urban and suburban settings in Greece [20], led us to the localized study of air pollution vs. thyroid status in pregnant women in Athens, Greece. We hypothesized that a dose—effect relationship might exist between air pollutants and thyroid parameters in pregnant women with various degrees of thyroid dysfunction.

## 2. Materials and Methods

We retrospectively evaluated thyroid function status (with thyrotropin—TSH) in 293 caucasian Greek women (mean age ± SD: 30.9 ± 5.9 years) in the second or third trimester of pregnancy vs. the average preceding nine-month NO_2_, PM2.5 and PM10 levels from five government-run air quality measuring stations in the metropolitan Athens area [21]. In this study, we used readily available data from the subjects’ medical records, history and laboratory results. The study was approved by our hospital’s scientific board/ethics committee (No 18/2019)

### 2.1. Study Population

The women studied were part of a larger sample of women that were referred (post-screening) for thyroid dysfunction during pregnancy in the largest state-run maternity hospital in Greece (this hospital caters to the needs of approximately 7000 pregnant women per year in an area where approximately 32,000 women give birth annually). All the women had no prior thyroid disease and were diagnosed with hypothyroidism for the first time during their pregnancy. Characteristics of the study group are given in Table 1.

We only included women with TSH >2.5 mIU/L or >3.0 mIU/L if the initial thyroid assessment was done in the first or the second trimester of pregnancy respectively.

### 2.2. Exposure Measurement

According to the Greek authorities, pollutants are measured continuously throughout the 24-h period. The response time of the automated analyzers is about one minute, that is, approximately every minute, each analyzer gives a value and average hourly pollution values are calculated. The methods used for NO_2_ and PM2.5/PM10 are with chemilluminescence and beta radiation attenuation, respectively. Quality control is done by the ISO-certified National Atmospheric Quality Reference Laboratory (www.ypeka.gr). Exposure to air pollutants (with nine-month mean ± SD for NO_2_, PM2.5 and PM10 at 38.2 ± 16.8 μg/m^3^, 17.4 ± 3.5 μg/m^3^ and 32.4 ± 8.1 μg/m^3^, respectively) for each woman was considered according to her place of residence within a reasonable linear distance (up to 4–5 km) from one of the five air quality measurement stations (Figure 1 and Figure 2).

### 2.3. Analysis

Statistical analysis of each variable (maternal age, weight change in pregnancy and of air pollutants: PM10, PM2.5 or NO_2_) vs. TSH was done with ordinary least squares regression (OLS-R) and quantile regression [22] (OLS-R and Q-R, with Gretl, v.2019d, http://gretl.sourceforge.net/index.html).

## 3. Results

Using OLS-R, a significant (positive) correlation for logTSH was found only with PM2.5 (with *R* = +0.13, *p* = 0.02) (Figure 3a).

Further analysis with Q-R showed that each incremental unit increase in PM2.5 increased mean logTSH (±SE) between +0.029 (0.001) to +0.025 (0.001) mIU/L (for the 10th to the 90th response quantile, *p* < 0.01) (this increase is equal to a mean ± SE increase in TSH of 1.069 ± 1.002 mIU/L to 1.059 ± 1.002 mIU/L) (Figure 1b). The other parameters (maternal age, weight change in pregnancy and PM10 and NO_2_ levels) had no significant effect on TSH.

## 4. Discussion

Our results do indeed show a dose—response relationship between PM2.5 and maternal TSH, with a more pronounced effect on the lower TSH levels of our study sample (these levels being slightly higher than the used cut-offs) and an attenuation in the higher TSH levels. The relationship of PM2.5 to logTSH in our study was very subtle; its slope was +0.029 to +0.025 (whereas, for example, the slope of free thyroxine to logTSH is reported to be approximately −0.35 [23]). The form of this association may denote a more important effect of PM2.5 at conditions of normal (or almost normal) thyroid gland status, whereas other factors may be more important at much higher TSH levels, particularly when the thyroid gland is compromised (such as in case of thyroiditis) and thyroid hormones are low. Thus, our results complement recent epidemiological works of maternal thyroid (dys)function versus PM2.5 levels [1]. Τhe mechanisms involved in the pathophysiologic actions of ambient air pollutants—particularly of PM2.5—may be their toxicity, induction of oxidative stress and/or of inflammation and mutagenicity; all are currently under investigation [24]. Although advancing maternal age and increased body weight gain in pregnancy are associated with increase in maternal TSH [25,26], no such effects of them on TSH were noted in our sample; we can only speculate that this was the result of the sample size (additionally, regarding body weight change the range of values was also narrow).

There are caveats to be considered in this study. We used only maternal TSH and not other thyroid function/autoimmunity parameters. Although TSH is considered to be sufficient for the screening of thyroid function in pregnancy [27], the effect—if any—of pollution on any other relevant thyroid parameters was not assessed. The choice of analysing with logTSH instead of TSH values was guided by the relevant scientific literature, since thyroid hormones—the main determinants of TSH secretion—are linked to the latter mainly with log-linear relationships [28]. Another caveat is that we did not use satellite air pollution data—as done in other studies [1], because these cover only daytime (on cloudless days) [29,30,31]; the recent environmental degradation in Greece was noted to occur during nighttime in the winter due to the in-city use of open wood burning fireplaces; wood burning is a very important pollution source worldwide [32,33]. In lieu, we used “traditional”—readily available—data from five air pollution measuring stations, a number which may be adequate for a metropolitan area of approximately of 400 km^2^ with 3.75 × 10^6^ people [34,35]; data from a few other stations were less accessible and/or absent for the time period that was assessed. Nevertheless, there is spatial heterogeneity in ambient air pollution levels. The use of the nearest monitors’ measurements to assess women’s air pollution exposure may have introduced substantial exposure misclassification; we acknowledge this shortcoming of our pilot study—this is a problem to be dealt with in future studies.

## Figures and Tables

**Figure 1 medsci-08-00019-f001:**
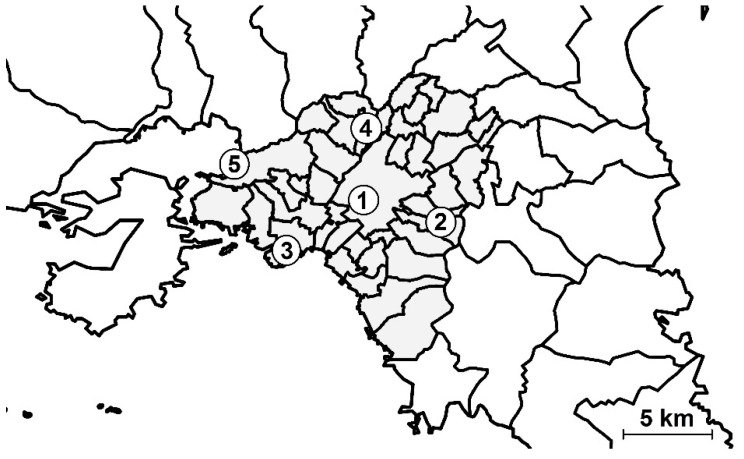
Air quality measurement stations in the metropolitan Athens area; 1: central Athens (urban), 2: AgiaParaskevi (suburban), 3: Pireaus (urban), 4: Lykovrisi (suburban), 5: Eleusis/Aspropyrgos (industrial); for each woman, the nine-month average of air pollutants’ levels from the closest station to her residence was taken into consideration.

**Figure 2 medsci-08-00019-f002:**
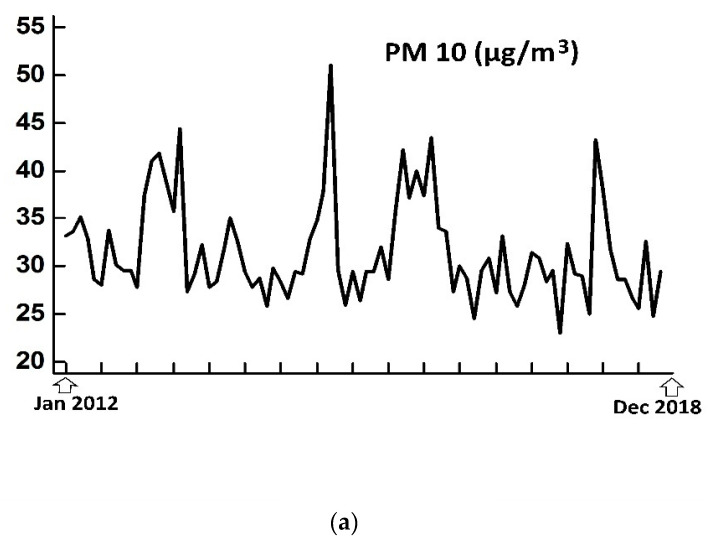
Average monthly air pollutants’ levels PM10 (**a**), PM2.5 (**b**) and NO_2_ (**c**) from the five monitoring stations) during the study period (January 2012 to December 2018).

**Figure 3 medsci-08-00019-f003:**
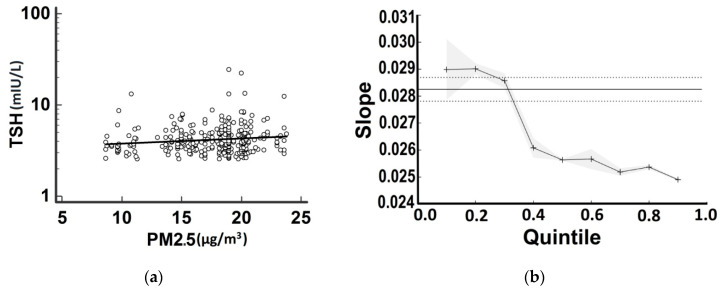
Scatter diagram with OLS-R for PM2.5 vs. logTSH (**a**) Graph of slopes for PM2.5 over logTSH (mIU/L) produced by Q-R (**b**) Note that the 0.5 quantile corresponds to the median and that the associations between each predictor are estimated to be stronger (over the levels estimated by OLS-R) at lower quantiles of response (i.e., TSH) and weaker (lower than the levels estimated by OLS-R) at higher quintiles of response. The straight line with dashed line borders indicates the OLS-R estimate with 95% confidence intervals, whereas the splined line with grey borders indicates the Q-R estimates with 95% confidence intervals.

**Table 1 medsci-08-00019-t001:** Characteristics of the study group.

Gestational age at inclusion (mean ± SD)	19.4 ± 8.6 weeks
Weight gain in pregnancy at inclusion (mean ± SD)	5.1 ± 5.6 kg
TSH at inclusion (mean ± SD)	4.54 ± 1.66 mIU/L

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
