# Peer review of "Atmospheric Pollution and Thyroid Function of Pregnant Women in Athens, Greece: A Pilot Study"

_medsci, 2020, doi:10.3390/medsci8020019_

Round 1

Reviewer 1 Report

This study aims to examine the association between air pollution exposure during pregnancy and thyroid function among pregnant women. While the topic is interesting, the manuscript is not well written with many important pieces missing. My major concerns are the sampling strategy, eligibility criteria, and exposure measurement. Below are my detailed comments.

  1. The introduction section can be expanded with more discussions on thyroid function as well as limitations of previous studies and how this study may address these gaps.
    2. Methods: please split this section into sub-sections to describe study population, exposure measurement, outcome measurement, covariates, and statistical analyses seperately.
    3. Methods: it is unclear how 293 women were sampled. Did the authors used convenient sampling strategy? More details are needed for the readers to understand how representative the results are.
    4. Methods: What is the rationale to only focus on women with hypothyroidism? Why not include women without hypothyroidism?
    5. There are large spatial heterogeneity in ambient air pollution levels. The authors used the nearest monitors' measurements to assess women's air pollution exposure which may introduce substantial exposure misclassification. Spatial interpolation such as kriging should be used to generate the exposure surface. Alternatively, more advanced exposure assessment models such as LUR or fused satelitte and monitored data can be used to improve the exposure assessment.
    6. There is no descriptions on what covariates were included. The authors mentioned that maternal age, weight change in pregnancy, and air pollutants were included in the model in the statistical analyses part. However, a seperated section to describe covariates (how they were measured and coded) is needed.
    7. How did the authors determine which confounders to adjust for in the model? Did they used DAG to guide the selection?
    8. The results lack a table 1 showing the characteristics of the women included in this study.

Author Response

Reviewer #1

We thank the reviewer for his/her comments – please see below how we dealt with them point by point.

[1]. The introduction section can be expanded with more discussions on thyroid function as well as limitations of previous studies and how this study may address these gaps.

We added a new paragraph as follows [ln 49 to ln 59]:

“The offspring of pregnant women that have been exposed to air pollutants show structural brain alterations, reduced executive function and behavioral problems. The tentative mechanisms for these problems may be oxidative stress, neuroinflammation or changes in the hypothalamic-pituitary-adrenal axis [1]. Thyroid hormones are crucial in the development of the fetal brain. The fetal thyroid gland is fully functional after mid-pregnancy, thus undiagnosed or less than optimally treated thyroid insufficiency early on in pregnancy adversely affects the development the offspring. Studies have shown a link between pollutants (polycyclic aromatic hydrocarbons) and maternal thyroid function [1]. Particulate air pollutants are prevalent and harmful to humans (as assessed versus mortality, morbidity or cardiovascular disease)[2]; however, data on whether exposure to particulate air pollutants can also disrupt thyroid thyroid function in pregnancy are limited.”

[2]. Methods: please split this section into sub-sections to describe study population, exposure measurement, outcome measurement, covariates, and statistical analyses seperately.

This section has been split in subsections as indicated

[3]. Methods: it is unclear how 293 women were sampled. Did the authors used convenient sampling strategy? More details are needed for the readers to understand how representative the results are.

We provide further details on the sample as follows [ln 80 to ln 92]

“… In this study we used readily available data from the subjects’ medical records, history and laboratory results.   2.1. Study populationThe women studied were part of a larger sample of women that were referred (post-screening) for thyroid dysfunction during pregnancy in the largest state-run maternity hospital in Greece (this hospital caters to the needs of approximately 7000 pregnant women per year in an area where approximately 32000 women give birth annually). All the women had no prior thyroid disease and were diagnosed with hypothyroidism for the first time during their pregnancy. Characteristics of the study group are given in Table 1. 

Table 1. Characteristics of the study group.

Gestational age at inclusion (mean+SD)

19.4±8.6 weeks

Weight gain in pregnancy at inclusion (mean+SD)

5.1±5.6 kg

TSH at inclusion (mean+SD)

4.54±1.66 mIU/L

[4]. Methods: What is the rationale to only focus on women with hypothyroidism? Why not include women without hypothyroidism?

These comprised a group of women that were referred for endocrine assessment post-screening – this is clarified as follows [ln 84 to ln 85]:

“The women studied were part of a larger sample of women that were referred (post-screening) for thyroid dysfunction during pregnancy”

[5]. There are large spatial heterogeneity in ambient air pollution levels. The authors used the nearest monitors' measurements to assess women's air pollution exposure which may introduce substantial exposure misclassification. Spatial interpolation such as kriging should be used to generate the exposure surface. Alternatively, more advanced exposure assessment models such as LUR or fused satelitte and monitored data can be used to improve the exposure assessment.

This is acknowledged in the Discussion section as follows [ln 177 to ln 180]

“there is spatial heterogeneity in ambient air pollution levels. The use of the nearest monitors' measurements to assess women's air pollution exposure may have introduced substantial exposure misclassification; we acknowledge this shortcoming of our pilot study - this is a problem to be dealt with in future studies.”

[6]. There is no descriptions on what covariates were included. The authors mentioned that maternal age, weight change in pregnancy, and air pollutants were included in the model in the statistical analyses part. However, a seperated section to describe covariates (how they were measured and coded) is needed.

In our analysis we used maternal age and maternal body weight change (as well as the pollutants’ levels) as variables (not covariates) against TSH – this is clarified in the Discussion section as follows [ln 160 to ln 162] and relevant references from the scientific literature are given:

“…advancing maternal age and increased body weight gain in pregnancy are associated with increase in maternal TSH…”

[7]. How did the authors determine which confounders to adjust for in the model? Did they used DAG to guide the selection?

Please see our answer above to point [6]

[8]. The results lack a table 1 showing the characteristics of the women included in this study.

A relevant table has been added moving data from the text to tabular form - Please see our answer above to point [3].

Reviewer 2 Report

The Authors present a short, but generally well-presented study of the impacts of air pollution on TSH levels in pregnant women living in Athens, Greece.  I only have a few comments that I would like the Authors to address or fix in the manuscript.

  1. The use of the word agglomeration does not really fit the context of the places that it is used.  Please use another.
  2. What seasons were the pregnancies examined?  Others have found that air pollution components differ with season and it might be useful for Readers to interpret the findings in this context.
  3. What is the relationship of the change in PM2.5 levels with changes to levels of TSH?
  4. The Authors should provide data values or figures for PM10 and NO2 to increase the impact of the paper.
  5. How was distance from air quality measurement stations included in the analyses?  Possibly a map showing location of the stations to where the women live.

Author Response

Reviewer #2

We thank the reviewer for his/her comments – please see below how we dealt with them point by point.

[1]. The use of the word agglomeration does not really fit the context of the places that it is used.  Please use another.

This has been changed to metropolitan area

[2]. What seasons were the pregnancies examined?  Others have found that air pollution components differ with season and it might be useful for Readers to interpret the findings in this context.

The subjects were included at various times throughout the years. Although seasonality is observed in air pollution data the analysis was done using the overall exposure over the averaged preceding nine months for each woman. 

[3]. What is the relationship of the change in PM2.5 levels with changes to levels of TSH?

This is detailed in the results section as follows [ln 142 to ln 145]:

“…Further analysis with Q-R showed that each incremental unit increase in PM2.5 increased mean logTSH (+SE) between +0.029 (0.001) to +0.025 (0.001) mIU/L (for the 10th to the 90th response quantile, p<0.01)(this increase is equal to a mean+SE increase in TSH of 1.069+1.002 mIU/L to 1.059+1.002 mIU/L)”

[4]. The Authors should provide data values or figures for PM10 and NO2 to increase the impact of the paper.

The relevant figures have been added to the revised manuscript.

[5]. How was distance from air quality measurement stations included in the analyses?  Possibly a map showing location of the stations to where the women live.

Linear distance was calculated (with Google maps) taking into account the air quality monitors’  location coordinates and the subjects’ given residence address. A figure indicating the approximate location of the air monitoring stations was also added.